# Improves the Resilience of Cucumber Seedlings under High-Light Stress through End-of-Day Addition of a Low Intensity of a Single Light Quality

Xue Li [1], Shiwen Zhao [1], Chun Qiu [1], Qianqian Cao [1], Peng Xu [1], Guanzhi Zhang [1], Yongjun Wu [2] and Zhenchao Yang [1,*]

[1] College of Horticulture, Northwest A & F University, Xianyang 712100, China
[2] College of Life Sciences, Northwest A & F University, Yangling 712100, China
* Correspondence: yangzhenchao@nwafu.edu.cn

**Abstract:** In order to investigate whether an end-of-day (EOD) addition of a single light quality could help alleviate high-light stress in a cucumber, cucumber seedlings were subjected to a 9 d period of high-light stress (light intensity was $1300 \pm 50$ $\mu mol \cdot m^{-2} \cdot s^{-1}$) when they were growing to 3 leaves and 1 heart, while the red light (R), blue light (B), green light (G), far-red light (FR), and ultraviolet A (UVA) light were added in the end-of-day period. The present study was conducted to measure antioxidants, chlorophyll content, and its synthetic degradative enzymes and chlorophyll a fluorescence in response to the degree of stress in cucumber seedlings. The experimental results demonstrated that the addition of blue light, UVA light, and green light significantly decreased the SOD and POD activities in the middle of the treatment (6th day) compared to the dark (D) treatment and improved the absorption performance of the PSI reaction centre of the cucumber seedling leaves to a certain extent ($PI_{ABS}$), but the PSII capacity capture ability (TRo/RC) of the three treatments decreased compared to the D treatment. The MDA content of all the treatments had a significant decrease compared to that of the D treatment. The MDA content of all the treatments was significantly lower than that of D, and its $F_V/F_M$ was increased to different degrees; the chlorophyll degrading enzyme PPH activity was significantly lower than that of the D treatment when a single light quality was added at the EOD period on the 9th d of treatment. In conclusion, cucumber seedlings subjected to short-term high-light stress can be added during the EOD period with a low-light intensity of a single R, G, B, or UVA light.

**Keywords:** ROS; photosynthetic efficiency; chlorophyll; high-light stress

## 1. Introduction

Plants require a solar light source for photosynthesis, but when the light intensity exceeds the optimal range required for photosynthesis, it will result in abiotic stress and physiological damage to the plant [1] and light can be one of the main abiotic stressors [2]. When plants are exposed to light stress, photoinhibition will be activated, leading to an uneven distribution of energy under PSI and PSII, which results in a decrease in their photosynthetic efficiency [3]. On the other hand, plants can mitigate the stress effects of excessive light intensity by reducing light absorption, limiting redox reactions, scavenging reactive oxygen species (ROS), and many other ways [4,5]. Plants have developed several photoprotective strategies to dissipate excess light energy. When exposed to higher light intensities, plants exhibit additional adaptation symptoms, including increased leaf thickness, reduced chlorophyll (Chl) content, and adjustments in chloroplast fine structure [6]. When the leaf captures more light energy than is required, it will induce the production of the ROS, which will directly inactivate the photochemical reaction centres of PSII, with a decrease in the PSII activity and a rapid decline in photosynthetic efficiency [7], and energy

dissipated in the form of fluorescence and heat will result in damage to the PSI and PSII reaction centres [8].

Light is not only used as an energy source to participate in plant photosynthesis but also as a signal source to regulate plant growth, differentiation, and metabolism [9]. Plants can receive light signals through different photoreceptors, which enable plants to complete corresponding responses under different light signals. Known plant photoreceptors include red/far-red light-absorbing phytochromes (PHY), blue and UVA receptors cryptochromes (CRY), phototropins (PHOT), and UVB-absorbing UVR8 [10]. In recent years, due to their controllability, artificial light sources have been widely used in the regulation of plant light environments [11]. Numerous researchers have investigated the effects of different wavelengths of light on plant growth and development [12,13]. However, less attention has been paid to end-of-light (EOD) and most of the existing studies on EOD have focused on the phenomenon that R/FR changes the morphology of photosensitive pigments (Pr/Pfr) to the extent that it affects the growth of plant hypocotyls [14–17], and little research has been conducted on the end-of-day additive effects of the remaining light qualities.

Light quality has a significant impact on plant morphology and secondary metabolite synthesis [18]. It has been shown that supplementation with UVA and UVB will lead to increased anthocyanin content in plants [19–21]. Blue light promotes the accumulation of photomorphogenesis and phototropism in plant leaves [22,23]. Red and far-red light activate plant photosensitive pigments affecting plant elongation growth [24], while green light reverses by blue light plant stomatal opening and promotes plant hypocotyl elongation [25]. Numerous results suggest that different monochromatic lights have different effects on plant growth and may also affect the ability of plants to cope with abiotic stresses (e.g., high light) [26].

The cucumber (*Cucumis sativus* L.) is widely cultivated worldwide as a broad-spectrum vegetable, but its seedlings are often subjected to abiotic stresses, including high-light stress when cultivated on land in summer. Different single light quality has a significant effect on the synthesis of plant secondary metabolites, and many secondary metabolites in plants play a key role in abiotic stress, different light quality has a significant effect on plant growth and photosynthesis, so this study uses the cucumber as the experimental material, to explore whether the addition of a single light quality before the darkness has a mitigating effect on cucumber subjected to high-light stress, and to provide a certain reference significance for the cultivation of cucumber seedlings in the greenhouse in the future.

## 2. Materials and Methods

### 2.1. Plant Materials

The experiment was conducted in 2022 at the Laboratory of Biological and Environmental Engineering for Facility Agriculture, College of Horticulture, Northwest A&F University. The material was a cucumber 'Xinjin You No. 1' (*Cucumis sativus* L), and the seeds were purchased from Huayi Seed Industry Co. in Tai'an, China. The cucumber seeds were germinated and sown in 10 cm × 10 cm nutrient pots and grown in an artificial climate chamber with a light intensity of 150 $\mu mol \cdot m^{-2} \cdot s^{-1}$, day and night temperatures of 25 °C/20 °C, photoperiods of 12 h/12 h, relative humidity of 60%, and water in moderation. At the time of growth to about 2 leaves and 1 heart, high light and single light quality irradiation was performed. They were watered with 1/4 Hoagland cucumber nutrient solution (pH 6.5 ± 0.1, EC 2.2–2.5 ms/cm), in which day and night temperatures were 28 °C/25 °C and relative humidity 40–50%, and the relevant indexes were measured on days 3, 6, and 9 of the treatment, respectively.

### 2.2. Treatments

When the cucumber seedlings grew to 2 leaves and 1 heart, they were irradiated with a sodium lamp with a light intensity of 1300 ± 50 $\mu mol \cdot m^{-2} \cdot s^{-1}$ during the daytime, and at the end of the daytime, they were irradiated with a single red light (R, 600–650 nm), with a light intensity of 50 $\mu mol \cdot m^{-2} \cdot s^{-1}$ for 1 h, respectively, with blue light (B, 450–500 nm),

green light (G, 500–550 nm), UVA (UVA, 320–400 nm), far-red light (FR, 700–750 nm), and darkness (D) as the control treatment. The LED light source adopts the LED lamp board and LED control system V1.0 produced by China Xi'an Inverter Optoelectronics Technology Co. The sodium light source is produced by China Basta Lighting Company(Jinhua, China).

### 2.3. Item Determination

#### 2.3.1. Malondialdehyde Content

The malondialdehyde (MDA) content was measured using a malondialdehyde content assay kit (Solebo, Beijing, China). The steps were as follows: weigh 0.1 g of tissue, add 1 mL of extraction solution for ice bath homogenisation, centrifuge at $8000 \times g$ for 10 min at 4 °C, take the supernatant, add the reagents sequentially according to the steps of the instruction manual, and 100 °C water bath for 60 min and then $10,000 \times g$. After centrifugation at room temperature for 10 min, the absorbance was measured at 532 nm and 600 nm, and the MDA content was calculated according to the instruction manual.

#### 2.3.2. Antioxidant Enzyme Activity

The catalase (CAT) activity was determined using a catalase activity assay kit (Solechem, Beijing, China); the superoxide dismutase (SOD) was determined using a superoxide dismutase activity assay kit (Solechem, Beijing, China); and the peroxidase (POD) content was determined using a peroxidase activity assay kit (Solechem, Beijing, China).

#### 2.3.3. Chlorophyll Content and Its Synthesis and Degradation Key Enzyme Activities

The 2nd true leaves of three cucumber seedlings from each treatment were selected and the content of chlorophyll a, chlorophyll b, and carotenoids was determined with reference to the method of Junfeng Gao [27].

$$C_a = 13.95 \times A_{665} - 6.88 \times A_{649} \tag{1}$$

$$C_b = 24.96 \times A_{649} - 7.32 \times A_{665} \tag{2}$$

$$C_{x \cdot c} = (1000 \times A_{470} - 2.05 \times C_a - 114.8 \times C_b)/245 \tag{3}$$

Note: In the formula $C_a$, $C_b$ represents the concentration of chlorophyll a and chlorophyll b, $C_{x \cdot c}$ represents the total concentration of carotenoids, and $A_{665}$, $A_{649}$, $A_{470}$ represents the absorbance of chloroplast pigment extract at wavelength 665 nm, 649 nm, 470 nm.

PAO enzyme activity in each treated plant was determined using a plant demagnesium chlorophyll a oxidase ELISA test kit (enzyme-free, Yancheng, China), PPH enzyme activity in each treated plant was determined using a plant demagnesium chlorophyllase ELISA test kit (enzyme-free, Yancheng, China), plant magnesium chelatase ELISA test kit (enzyme-free, Yancheng, China), and plant magnesium chelatase (enzyme-free, Yancheng, China). The MgCH enzyme activity was determined in each treated plant, and the FeCH enzyme activity was determined in each treated plant using a plant ferrous chelatase ELISA kit (enzyme-free, Yancheng, China).

#### 2.3.4. Determination of Parameters Related to Plant Photosynthetic Efficiency

The 2nd true leaves of three cucumber seedlings from each treatment were selected on the 9th day of the treatments carried out, and the transpiration rate (E), net photosynthetic rate (Pn), intercellular $CO_2$ concentration (Ci), and stomatal conductance (gsw) were measured using the Plant Photosynthesis Tester 6800 (LI-6800, Lincoln, NE, USA). The leaf chamber temperature was set at 24 °C, the $CO_2$ level at 400 μmol/mol, the relative humidity at 60%, and the light source for the determination was set to be R90B10 with a light intensity of 1000 $μmol·m^{-2}·s^{-1}$.

### 2.3.5. Determination of Chlorophyll a Fluorescence Parameters

The 2nd true leaves of three cucumber seedlings from each treatment were selected on the 9th day of treatment, and the chlorophyll fluorescence fast kinetic curves (OJIP curves), related parameters, and 820 nm light reflectance curves were measured on leaves dark-adapted to the treatments for 60 min by using a plant efficiency meter (M-PEA, UK). The parameters include the PSII performance index based on absorption ($PI_{ABS}$), PSII maximum photochemical efficiency ($F_V/F_M$), relative fluorescence intensity at 300 μs (Vk), relative fluorescence intensity at 2 ms (Vj), and relative fluorescence intensity at 30 ms (Vi). The significance represented by each parameter is shown in Table 1.

**Table 1.** Abbreviations and interpretation of parameters related to chlorophyll a fluorescence.

| Chlorophyll a Fluorescence Parameters | An Explanation of the Meaning of Words or Phrases |
|---|---|
| $F_V/F_M$ | It represents the maximum quantum yield of PSII |
| $F_V/F_O$ | It represents the maximum efficiency of the water-splitting complex |
| $S_M = Area/(F_M - Fo)$ | It represents the multiple turnovers of $Q_A$ reductions |
| $V_J$ | Relative variable fluorescence at phase J of the fluorescence induction curve |
| $V_I$ | Relative variable fluorescence at phase I of the fluorescence induction curve |
| $PI_{ABS} = \gamma RC/(1 - \gamma RC) \times \varphi Po/(1 - \varphi Po) \times \psi o/(1 - \psi o)$ | Performance index of PS I on absorption basis |
| $PI_{TOTAL} = PI_{ABS} \times \delta Ro/(1 - \delta Ro)$ | Performance index of electron flux to the final PS I electron acceptors |
| $\varphi Po$ | Maximum quantum yield of primary PSII photochemistry (at t = 0) |
| $\varphi(Eo)$ | Quantum yield (at t = 0) for electron transport from QA- to plastoquinone |
| $\psi o$ | Probability (at t = 0) that a trapped exciton moves an electron into the electron transport chain beyond QA |
| YRC | The probability that PSII chlorophyll molecule functions as RC |
| $\delta(Ro) = (1 - VJ)/(1 - VI)$ | Efficiency/probability (at t = 0) with which an electron from the intersystem carriers moves to reduce end electron acceptors at the PSI acceptor side |
| $ABS/RC = (1 - \gamma RC)/\gamma RC$ | Absorption flux per RC corresponding directly to its apparent antenna size |
| $TRo/RC = \Delta V/\Delta t0 \times (1/Vj)$ | Trapping flux leading to QA reduction per RC at t = 0 |
| $ETo/RC = \Delta V/\Delta t0 \times (1/Vj) \psi 0$ | Electron transport flux from QA- to plastoquinone per RC at t = 0 |
| $DIo/RC = (ABS/RC - TR0/RC)$ | Dissipated energy flux per RC at the initial moment of the measurement, i.e., at t = 0 |
| $ABS/RC = (1 - \gamma RC)/\gamma RC$ | Absorption flux per RC corresponding directly to its apparent antenna size |
| ABS/CSm | Absorption of energy per excited cross-section (CS) approximated by $F_M$ |
| TRo/CSm | Excitation energy flux trapped by PSII of a Photosynthesising sample cross-section (CS) approximated by $F_M$ |
| ETo/CSm | Electron flux transported by PSII of a |

### 2.3.6. Determination of Secondary Metabolites

The 2nd true leaves of three cucumber seedlings from each treatment were selected on the 9th day of the treatments carried out, and the samples were taken and fully ground for the determination of flavonoid content using a flavonoid content assay kit (Solebo, Beijing, China) and the total phenol content using a total phenol content assay kit (Solebo, Beijing, China).

### 2.4. Statistical Analyses

Data were collated using Microsoft Office Excel 2020, IBM SPSS Statistics 25 significant difference analysis, and GraphPad Prism 9.5 plotting. The results are presented as the mean values ± standard errors. The means with the same letters in different columns are not significantly different and the mean difference was determined using Duncan's multiple range test (DMRT) at $p \leq 0.05$.

## 3. Results

### 3.1. Effects of EOD Addition of Different Single Light Qualities before Dark on Antioxidant Content of Cucumber Seedling Leaves under High-Light Stress

We determined a number of parameters including MDA, $H_2O_2$, SOD, and POD were measured on the 3rd, 6th, and 9th day under different treatments and the results were obtained as shown in Figure 1. As shown in Figure 1a, the MDA content under R, G and UVA treatments was significantly lower than that of D treatment at 3d, and at 6d and 9d, the MDA content under each treatment was lower than that of D treatment. As shown in Figure 1b, the CAT activity under B, G, and UVA treatments was significantly lower than the rest of the treatment groups on the 9th day. As shown in Figure 1c, the SOD activity under B, G, and UVA treatments was significantly lower than the rest of the treatments on the 9th day. As shown in Figure 1d, the POD activity under D treatment was significantly higher than the rest of the treatments on the 6th day and 9th day.

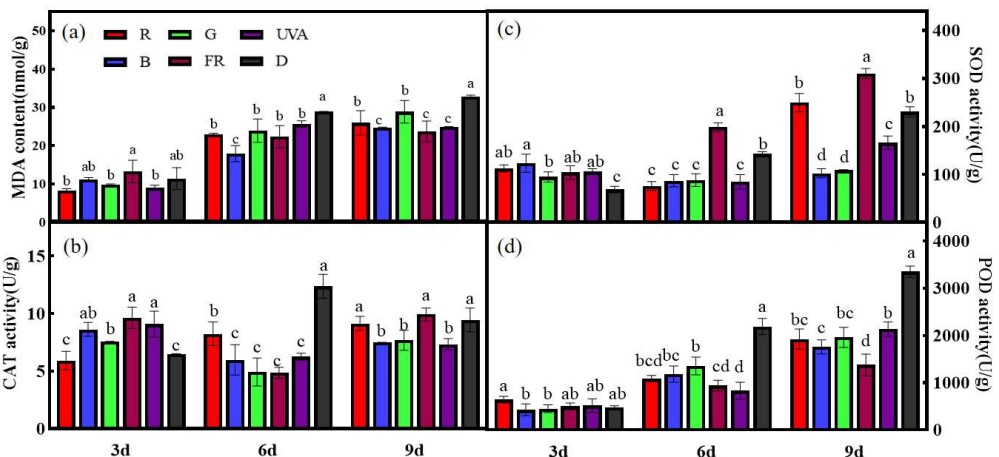

**Figure 1.** The antioxidant content of cucumber seedlings under different treatments ((**a**) the MDA content of cucumber seedlings under different treatments, (**b**) the CAT activity of cucumber seedlings under different treatments, (**c**) the SOD content of cucumber seedlings under different treatments, and (**d**) the POD content of cucumber seedlings under different treatments). Note: Where R stands for red light treatment during EOD, B stands for blue light treatment during EOD, G stands for green light treatment during EOD, UVA stands for UVA light treatment during EOD, and D stands for dark treatment during EOD, where D is the control. The 3d, 6d, and 9d on the x-axis represent different treatment days. The above comparisons of differences were made only among different treatments at the same time period. The mean difference was determined using Duncan's multiple range test (DMRT) at $p \leq 0.05$. The abbreviated letters in the following figure represent the same meaning as the treatments.

### 3.2. Effects of EOD Addition of Different Single Light Qualities before Dark on Chlorophyll Content of Cucumber Seedlings under High-Light Stress

The chlorophyll content was determined under different treatments on the 3rd, 6th, and 9th day. As shown in Figure 2a, the chlorophyll a content under UVA and D treatments was significantly lower than the rest of the treatments on the 6th day, but there was no significant difference in the chlorophyll a content among the treatments on the 9th day. As shown in Figure 2b, the chlorophyll b content was significantly higher under G, FR, and UVA treatments than the rest of the treatments on the 9th day. The carotenoid content was significantly higher under R and G treatment than the rest of the treatments on the 6th day and significantly lower under D, FR, and UVA treatments than the rest of the treatments on the 9th day.

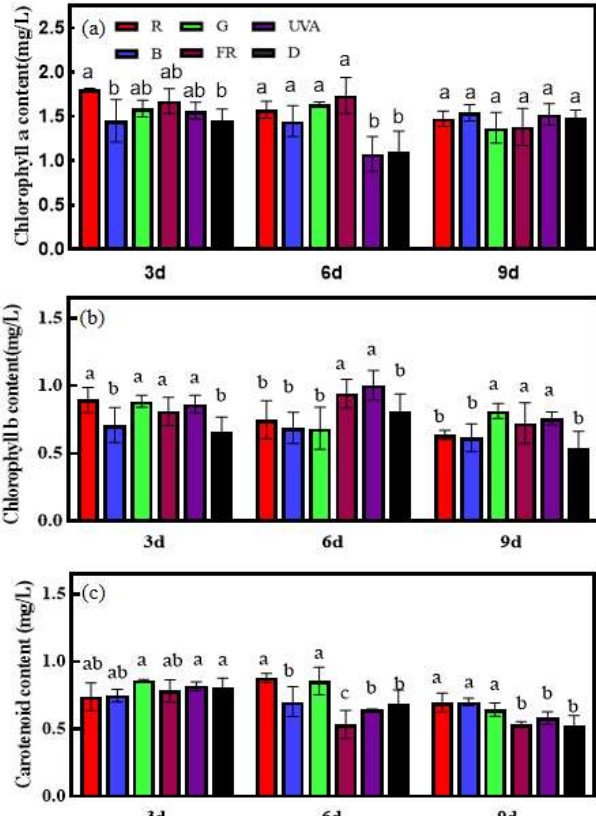

**Figure 2.** The chlorophyll content of cucumber seedlings under different treatments ((**a**) Chlorophyll a content of cucumber seedlings under different treatments, (**b**) Chlorophyll b content of cucumber seedlings under different treatments, and (**c**) Carotenoid content of cucumber seedlings under different treatments). The mean difference was determined using Duncan's multiple range test (DMRT) at $p \leq 0.05$.

### 3.3. Effects of EOD Addition of Different Single Light Qualities before Dark on Chlorophyll-Related Enzyme Activities in Cucumber Seedlings under High-Light Stress

For chlorophyll degradation key enzyme activities PPH and PAO, chlorophyll synthesis key enzymes MgCH and FeCH activities were determined, and the results were obtained as shown in Figure 3. As shown in Figure 3a, the PPH enzyme activity under the D treatment was significantly higher than the rest of the treatments at 6d and 9d of the treatments. As shown in Figure 3b, the PAO activity under the FR treatment was lower than the D on the 3rd day and 6th day, but there was no significant change in the PAO activity among treatments on the 9th day. As shown in Figure 3c, the addition of G increased the chlorophyll synthase MgCH activity on the 6th day and 9th day. As shown in Figure 3d, the FeCH activity was higher than the D treatment with the addition of R, B, G, FR, and UVA on the 3rd day. The FeCH activity was significantly higher under the FR treatment than under the D treatment on the 6th day and 9th day, and enzyme activity was significantly higher under the G and FR treatments than the D treatment on the 9th day.

### 3.4. Effects of EOD Addition of Different Single Light Qualities before Dark on Photosynthesis of Cucumber Seedlings under High-Light Stress

For the cucumber seedlings of treatment, 9th-day photosynthesis-related parameters were determined to obtain the results shown in Figure 4. As shown in Figure 4a, the net photosynthetic rate Pn was significantly higher in the treatments with the addition of B, G, and UVA than in the D treatment, but the Pn was significantly lower under the FR treatment than in the rest of the treatments. As obtained from Figure 4c,d, the stomatal conductance gsw and transpiration rate E were significantly higher under the UVA and D treatments than the remaining treatments.

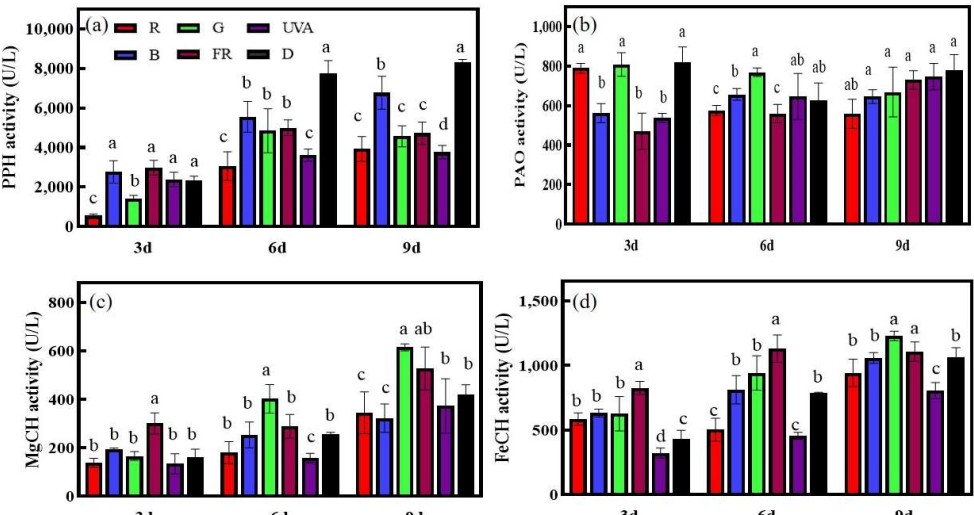

**Figure 3.** Chlorophyll synthesis and degradation enzyme activities in cucumber seedlings under different treatments ((**a**) PPH activity in cucumber seedlings under different treatments, (**b**) PAO activity in cucumber seedlings under different treatments, (**c**) MgCH activity in cucumber seedlings under different treatments, (**d**) and FeCH activity in cucumber seedlings under different treatments). The mean difference was determined using Duncan's multiple range test (DMRT) at $p \leq 0.05$.

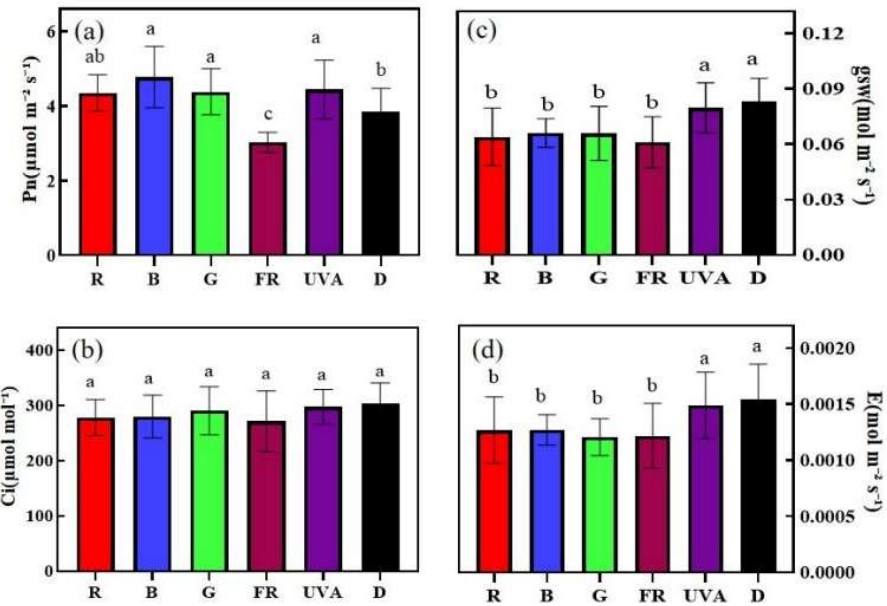

**Figure 4.** Chlorophyll synthesis and degradation enzyme activities of cucumber seedlings under different treatments ((**a**) net photosynthetic rate of cucumber seedlings under different treatments, (**b**) intercellular carbon dioxide of cucumber seedlings under different treatments, (**c**) stomatal conductance of cucumber seedlings under different treatments, (**d**) and transpiration rate of cucumber seedlings under different treatments). The mean difference was determined using Duncan's multiple range test (DMRT) at $p \leq 0.05$.

*3.5. Effects of EOD Addition of Different Single Light Qualities on Chlorophyll a Fluorescence Parameters and Photosystem Energy Conversion Efficiency in Cucumber Seedlings under High-Light Stress*

The chlorophyll a fluorescence parameters of cucumber seedlings on the 9th day of treatment were determined and the JIP-related data were shown in Figure 5 and Table 2. The $F_V/F_M$ values were significantly increased by adding G, B and UVA before darkness compared with that of D. The SM values were significantly decreased under the G and FR

treatments compared with the rest of the treatments. There was no significant difference in $V_J$ among treatments, but the value of VI was significantly higher in G compared with B and UVA, and there was no significant difference among the remaining treatments. The $PI_{ABS}$ was significantly higher in the R treatment than in the FR and D treatments, and the $PI_{ABS}$ was somewhat higher in all treatments (except FR) than in the D treatment, but there was no significant difference. The $PI_{TOTAL}$ was significantly higher in the R, B and UVA treatments than in the rest of the treatments.

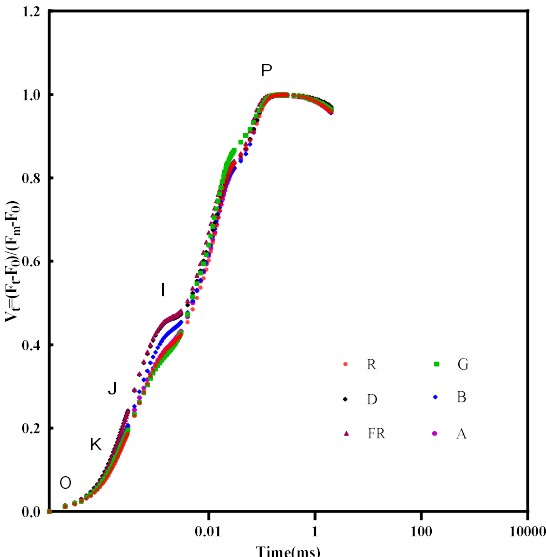

**Figure 5.** OJIP curves of cucumber seedlings under different treatments.

**Table 2.** Other JIP Parameters under different treatment.

| Treatments | $F_V/F_M$ | $F_V/F_O$ | $S_M$ | $V_J$ | $V_I$ | $PI_{ABS}$ | $PI_{TOTAL}$ |
|---|---|---|---|---|---|---|---|
| R | $0.805 \pm 0.002$ ab | $4.404 \pm 0.059$ a | $17.932 \pm 1.331$ a | $0.404 \pm 0.053$ a | $0.837 \pm 0.009$ ab | $3.394 \pm 0.584$ a | $1.272 \pm 0.166$ a |
| G | $0.811 \pm 0.007$ a | $4.285 \pm 0.178$ a | $15.139 \pm 1.387$ bc | $0.394 \pm 0.013$ a | $0.868 \pm 0.013$ a | $3.185 \pm 0.218$ ab | $0.891 \pm 0.099$ b |
| FR | $0.806 \pm 0.005$ ab | $4.18 \pm 0.399$ a | $14.457 \pm 1.165$ c | $0.471 \pm 0.017$ a | $0.843 \pm 0.028$ ab | $2.16 \pm 0.153$ b | $0.923 \pm 0.039$ b |
| D | $0.798 \pm 0.006$ b | $4.19 \pm 0.146$ a | $17.915 \pm 0.316$ a | $0.467 \pm 0.001$ a | $0.836 \pm 0.001$ ab | $2.171 \pm 0.147$ b | $0.967 \pm 0.064$ b |
| B | $0.815 \pm 0.009$ a | $4.409 \pm 0.246$ a | $17.574 \pm 2.448$ ab | $0.438 \pm 0.083$ a | $0.825 \pm 0.026$ b | $3.013 \pm 1.091$ ab | $1.365 \pm 0.521$ a |
| UVA | $0.813 \pm 0.007$ a | $4.355 \pm 0.201$ a | $17.737 \pm 0.715$ a | $0.408 \pm 0.034$ a | $0.825 \pm 0.027$ b | $3.054 \pm 0.529$ ab | $1.28 \pm 0.214$ a |

Note: Within the same column, different letters represent significant differences ($p < 0.05$), while the same letters represent no significant differences ($p > 0.05$). The same below.

The yield parameters under the different treatments on the 9th day are shown in Table 3, with no significant differences between treatments φPo, φ(Eo), ψo, and δ(Ro), and only the YRC treatments showed significant differences, with the G treatment having significantly lower YRC values than the remaining treatments.

**Table 3.** Yield Parameters under different treatments.

| Treatments | φPo | φ(Eo) | ψo | YRC | δ(Ro) |
|---|---|---|---|---|---|
| R | $0.815 \pm 0.002$ a | $0.486 \pm 0.043$ a | $0.596 \pm 0.053$ a | $0.133 \pm 0.007$ ab | $0.274 \pm 0.01$ a |
| G | $0.811 \pm 0.006$ a | $0.491 \pm 0.008$ a | $0.606 \pm 0.013$ a | $0.107 \pm 0.01$ b | $0.219 \pm 0.024$ a |
| FR | $0.806 \pm 0.015$ a | $0.426 \pm 0.011$ a | $0.529 \pm 0.017$ a | $0.127 \pm 0.022$ ab | $0.296 \pm 0.045$ a |
| D | $0.807 \pm 0.005$ a | $0.431 \pm 0.002$ a | $0.533 \pm 0.001$ a | $0.133 \pm 0.001$ ab | $0.308 \pm 0.001$ a |
| B | $0.815 \pm 0.009$ a | $0.457 \pm 0.066$ a | $0.562 \pm 0.083$ a | $0.142 \pm 0.021$ a | $0.311 \pm 0.009$ a |
| UVA | $0.813 \pm 0.007$ a | $0.482 \pm 0.031$ a | $0.592 \pm 0.034$ a | $0.142 \pm 0.021$ a | $0.298 \pm 0.065$ a |

Note: Within the same column, different letters represent significant differences ($p < 0.05$), while the same letters represent no significant differences ($p > 0.05$). The same below.

The specific energy flux under different treatments on the 9th day is shown in Table 4. The ABS/RC values were significantly lower under the R and B treatments than under

the FR and D treatments. The TRo/RC was significantly lower under the R, G, B, and UVA treatments than D. There was no significant difference in the Eto/RC values among treatments. The DIo/RC was significantly lower under the R and B treatments than the FR and D treatments.

**Table 4.** Specific energy flux under different treatments.

| Treatments | ABS/RC | TRo/RC | ETo/RC | DIo/RC |
|---|---|---|---|---|
| R | 1.944 ± 0.071 b | 1.585 ± 0.059 c | 0.947 ± 0.114 a | 0.360 ± 0.013 b |
| G | 2.073 ± 0.126 ab | 1.681 ± 0.115 bc | 1.018 ± 0.055 a | 0.392 ± 0.011 ab |
| FR | 2.171 ± 0.040 a | 1.750 ± 0.064 ab | 0.925 ± 0.023 a | 0.420 ± 0.026 a |
| D | 2.212 ± 0.079 a | 1.785 ± 0.052 a | 0.952 ± 0.029 a | 0.427 ± 0.027 a |
| B | 1.971 ± 0.157 b | 1.606 ± 0.122 bc | 0.897 ± 0.105 a | 0.365 ± 0.039 b |
| UVA | 2.094 ± 0.074 ab | 1.602 ± 0.058 bc | 1.009 ± 0.084 a | 0.391 ± 0.022 ab |

Note: Within the same column, different letters represent significant differences ($p < 0.05$), while the same letters represent no significant differences ($p > 0.05$). The same below.

Phenomenological Energy Flux under different treatments on the 9th day is shown in Table 5, where there was no significant difference between TRo/CSm and DIo/CSm for each treatment. The values of ABS/CSm and ETo/CSm under D treatment were significantly lower than those under UVA treatment, and there was no significant difference between the rest of the treatments.

**Table 5.** Phenomenological Energy Flux under different treatments.

| Treatments | ABS/CSm | TRo/CSm | ETo/CSm | DIo/CSm |
|---|---|---|---|---|
| R | 46,167.333 ± 2495.077 ab | 37,622.667 ± 2032.765 a | 22,508.333 ± 3124.695 ab | 8544.667 ± 474.031 a |
| G | 46,075.333 ± 2686.795 ab | 37,361.000 ± 2471.864 a | 22,620.333 ± 1154.449 ab | 8714.333 ± 222.194 a |
| FR | 44,793.667 ± 2660.432 ab | 36,137.000 ± 2811.193 a | 19,093.667 ± 1105.913 ab | 8656.667 ± 204.270 a |
| D | 43,224.000 ± 619.000 b | 34,892.000 ± 733.000 a | 18,613.000 ± 369.000 b | 8332.000 ± 114.000 a |
| B | 46,007.667 ± 1062.27 ab | 37,486.000 ± 715.199 a | 21,022.000 ± 2840.544 ab | 8521.667 ± 538.000 a |
| UVA | 47,812.333 ± 2680.798 a | 38,887.333 ± 2508.309 a | 23,084.000 ± 2732.314 a | 8925.000 ± 195.049 a |

Note: Within the same column, different letters represent significant differences ($p < 0.05$), while the same letters represent no significant differences ($p > 0.05$). The same below.

*3.6. Effect of Adding Different Single Light Qualities before Darkness on Flavonoids and Total Phenol Content of Cucumber Seedling Leaves under High-Light Stress*

For the determination of flavonoid content as shown in Figure 6a, the flavonoid content under B and UVA treatments was significantly higher than the rest of the treatments, the content under G light treatment was significantly higher than the D treatment, and there was no significant difference between the flavonoid content under R and FR treatments and the D treatment. The total phenolic content under each treatment is shown in Figure 6b, which was significantly higher under B and UVA treatments than the rest of the treatments, significantly lower under G treatment than the rest of the treatments, and not significantly different under R, FR, and D treatments.

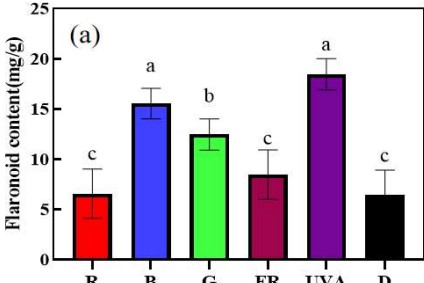 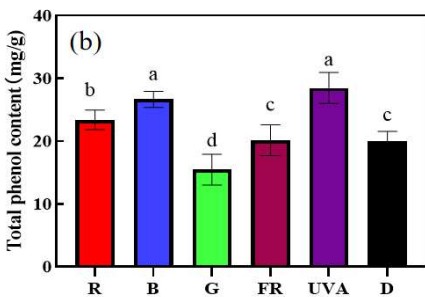

**Figure 6.** Flavonoid and total phenolic content of cucumber seedlings under different treatments. ((**a**). Flavonoid content of cucumber seedlings under different treatments, (**b**). Total phenol content

of cucumber seedlings under different treatments). The mean difference was determined using Duncan's multiple range test (DMRT) at $p \leq 0.05$.

## 4. Discussion

### 4.1. Different EOD Light Quality Affects Antioxidant and Secondary Metabolite Contents of Cucumber Seedlings

Light significantly affects the synthesis and accumulation of secondary metabolites critical for plant growth [18]. The ultraviolet (UV) absorption properties of flavonoids allow them to exert photoprotective properties [28], and in this experiment, UVA was added before darkness, and the flavonoid content of cucumber leaves was significantly higher under the B and G light treatments than under D. Rodriguez et al. [19] demonstrated that the addition of a low-dose UVB light under the HPSL resulted in an increase in the concentration of flavonoids in lettuce, which is the same as the results obtained in the present experiment with the addition of UVA at the EOD period. It has been shown that when the PPFD is constant, less blue light produces lower levels of flavonoids [29], so the blue light content is positively correlated with the flavonoid content. Phenolic compounds are one of the important defence mechanisms when organisms are subjected to periods of abiotic stress [30]. The total phenolic content is significantly affected by the light quality, and it has been shown that LED blue light significantly increases total phenolic and flavonoid content in tomato leaves [31]. Mohanty [32] and others showed that both red and blue light stimulated the synthesis of phenolic compounds in rice leaves. In the present experiment, the total phenolic content was significantly higher under EOD-added R, B and UVA treatments than D treatment, but significantly lower under G treatment compared to D treatment.

Light stress will lead to oxidative damage in plants. Chloroplast ROS are considered to be harmful substances of oxidative metabolism and are major regulatory substances in plants [33]. The leaf MDA content was determined in this study in response to the degree of cucumber stress under each treatment. On the 6th–9th day, both EOD additions of single light quality resulted in significantly lower MDA content in leaves than in the D treatment. It is evident that the addition of a single light quality before darkness has a mitigating effect on high light-induced stress. The biological reactive oxygen defence system consists of SOD, POD, and CAT. This system plays a role in preventing or reducing the form of hydroxyl radicals and eliminating superoxide radicals $H_2O_2$ and peroxides [34–37]. The most significant change was in the POD enzyme activity, which was lower than that of the D treatment in both treatments at 6d and 9d. Numerous studies have shown that the addition of B and R light to white light can promote the accumulation of antioxidants in plants [38–40]. The similar results obtained with the addition of G, UVA, and FR may be due to the promotion of the synthesis of secondary metabolites in cucumber seedlings in response to abiotic stresses.

### 4.2. Different EOD Light Quality Affects Chlorophyll Content and Synthase and Degradative Enzyme Activities in Cucumber Seedlings

Plants grown under excessive light will have fewer photosynthetic pigments than plants grown under suitable light [41]. At the end of treatment on the 9th day, there was no significant difference in chlorophyll a content between groups, while chlorophyll b content was significantly higher in EOD plus G, FR, and UVA light treatments than the rest of the treatments, which may be due to the fact that the addition of single light quality by EOD played a catalytic role in the synthesis of secondary metabolites and antioxidants in the cucumber seedlings to enhance the resilience of the plant to abiotic stresses. The carotenoids in EOD-added R, G, and B were significantly higher than those in D treatment after the 9th day of treatment. It has been shown that the addition of green light will promote the accumulation of carotenoids [42,43]. It has also been shown that increasing the proportion of red light in the red–blue light mixture will promote carotenoid accumulation [44], but the results of whether blue light promotes the accumulation of carotenoids are inconsistent due

to species differences. Kyriacou et al. [45] demonstrated that supplementation with a single blue light reduced the carotenoid content of Amaranthus, watercress, and Amaranthus plants. Ouzounis et al. [23] determined that the supplementation of LED blue light in daylight would promote carotenoid accumulation in lettuce. The chlorophyll degradation and synthesis involves several enzymes. PAO is involved in the first part of the chlorophyll catabolism, and the PAO/phyllobilin pathway is responsible for chlorophyll degradation after the "chlorophyll cycle" [46]. According to previous studies, the PPH has been shown to be essential for chlorophyll catabolism during leaf brushing in a variety of plants, while FeCH has been shown to be essential for chlorophyll catabolism during the leaf brushing period [47–49], and FeCH is required for chlorophyll catabolism during the leaf brushing period, while FeCH and MgCH are key enzymes for chlorophyll synthesis [50]. From the present experiment, the addition of single light quality during the EOD period significantly reduced the PPH enzyme activity on the 6th day and 9th day of the treatments, whereas there was no significant difference in PAO activity among the treatments on the 9th day period. For the chlorophyll synthase MgCH and FeCH enzyme activities, there were differences among the treatments; the G and FR treatments showed a significant increase in MgCH and FeCH enzyme activities compared to the D treatment on the 9th day of the treatment.

### 4.3. Different EOD Light Quality Affects Photosynthesis and Chlorophyll Fluorescence of Cucumber Seedlings

Plant photosynthesis is significantly regulated by the light environment it is exposed to. The present study showed that the EOD addition of B, G, and UVA significantly increased the net photosynthetic rate of cucumber seedlings under high-light stress compared to the D treatment, while the EOD addition of R increased but did not differ significantly from the D treatment. This is consistent with the study of Skarleth et al. [51] on lettuce, which showed that lettuce under light treatments of R or B added during the EOD period had higher photosynthetic capacity and photosynthetic potential than the control group. The effect of B added during the EOD period on lettuce was more carefully investigated by Viktorija et al. [52], but data collected during their main photoperiods similarly confirmed that the photosynthetic capacity of lettuce was significantly higher than that of the control group under the addition of B added during the EOD period. Lettuce photosynthetic capacity was significantly higher than that of the control group. Zou et al. [53] added FR during the EOD period to investigate its effect on lettuce, and the results showed that the net photosynthetic rate of FR-treated lettuce was significantly lower than that of the control group at a PPFD of more than 300 $\mu mol \cdot m^{-2} \cdot s^{-1}$, and showed a higher NPQ value at a PPFD of 600 $\mu mol \cdot m^{-2} \cdot s^{-1}$.

The $F_V/F_M$ values of the cucumber seedlings were significantly higher under all treatments except the FR treatment, which also responded to the increase in the level of stress to which the plants were subjected [54]. When the value was lower than 0.83, it indicated that the plants suffered from stress, and the smaller the value was compared to 0.83 indicated that they suffered from a deeper degree of stress, so the cucumber seedlings suffered from high-light stress under all treatments were alleviated to a certain extent. The photoinhibitory effect is usually perceived when light energy exceeds the photosynthetic capacity. During high-light stress, a decrease in quantum efficiency and the photosynthetic rate was observed, followed by an impairment of the photosynthetic apparatus, leading to functional failure and increased heat dissipation in the PSII reaction centres [55–57]. The $PI_{ABS}$ and $PI_{TOTAL}$ denote plant photosynthetic performance [58], and it has been shown that PIABS is a major contributing factor to the effect of plants subjected to high-light stress. The PSII reaction active centre is one of the most important parameters [59], and the results of this experiment concluded that the addition of a single light quality during the EOD period had a different degree of increase in PIABS compared to the D treatment, except for FR. The specific energy flux varied significantly under different treatments, with a significant decrease in absorbed flux (ABS/RC) and maximum rate of quantum capture

(TRo/RC) in both the R and B treatments compared to D. The results of this experiment showed that the PIABS increased significantly with the addition of a single photoplasma during the EOD period, except for FR. It has been shown that UV radiation primarily damaged the D1/D2 reaction centre proteins, the oxygen-evolving complex, and other components on both the acceptor and donor sides of PSII. In contrast to the PSII, the PSI activity in the rice seedlings was greatly enhanced by the UVB exposure [60,61]; B and UVA treatments reduced TRo/RC, indicating that treatment with a single light quality (R, B, G, UVA) during the EOD period reduced its photo-quantum capture efficiency under high-light stress, resulting in a reduction in the degree of high-light stress.

## 5. Conclusions

The addition of a single R, G, B, and UVA during the EOD period promoted the production of more secondary metabolites and reduced the light quantum capture capacity of the cucumber to cope with high-light stress. The addition of a low-dose single photoplasmas during the EOD period had a certain degree of alleviation of high-light stress in the cucumber seedlings: the cucumber maintained high photosynthetic capacity at the end of the treatment period (9th day) and the chlorophyll degradation and synthesis in the leaves remained at a normal level. In conclusion, cucumber seedlings subjected to short-term high-light stress can be added during the EOD period with a single R, G, B, or UVA light.

**Author Contributions:** Conceptualization, X.L.; methodology, X.L.; validation, X.L. and S.Z.; formal analysis, C.Q.; investigation, X.L. and S.Z.; data curation, X.L. and S.Z.; writing—original draft preparation, X.L.; writing—review and editing, Y.W. and Z.Y.; visualization, Q.C., G.Z. and P.X.; supervision, Z.Y. All authors have read and agreed to the published version of the manuscript.

**Funding:** Shaanxi Provincial Technological Innovation Guiding Special Project (2021QFY08-02); Shaanxi Province 100 Billion Facility Agriculture Special Project in 2021(K3030821094); Key Technological Innovation and Integration of Facility Vegetables in the Tibetan Plateau (XZ202202YD0002C); Introduction of Famous Varieties of Facility Vegetables, Melons and Fruits and Construction of Standardised Demonstration Bases (QYXTZX-AL2023-07).

**Institutional Review Board Statement:** Not applicable.

**Informed Consent Statement:** Not applicable.

**Data Availability Statement:** Data sharing not applicable.

**Conflicts of Interest:** The authors declare no conflict of interest.

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
