# Peer review of "Improves the Resilience of Cucumber Seedlings under High-Light Stress through End-of-Day Addition of a Low Intensity of a Single Light Quality"

_horticulturae, doi:10.3390/horticulturae9111237_

Round 1

Reviewer 1 Report

Comments and Suggestions for Authors

The manuscript describes the results of a study of additional irradiation of cucumber seedlings with LEDs at the end of the day. The manuscript is well structured and the conclusions are supported by the results. The results may be useful when growing seedlings under artificial light. I have a few questions. 1. Why was a light intensity of 1000 µmol m-2 s-1 chosen when measuring photosynthesis parameters? After all, the plants were grown at a light intensity of 1300±50 µmol m-2 s-1. 2. There is no reference to Table 1 in the text. 3. In conclusion, recommendations should be added on the use of additional irradiation when growing cucumber seedlings.

Reviewer 2 Report

Comments and Suggestions for Authors

The manuscript with the title End-of-day addition of a low intensity of a single light quality improves the resilience of cucumber seedlings under high light stress” is has all the parts of an article. I wonder why is it in the category of „Data Descriptor”?

The title could be reformulated.

The introduction section should be reformulated. For example „row 35 ROS” and „row 38 Chl content”-when first use an abreviation it should be reccomended to write the entire words only when first appeares in the text, then you can use the abbreviation in further explanations.

For example: Improved resilience of cucumber seedlings under end-of-day low intensity light

Row 151: The sentence is better to begin with „The parameters” MDA, etc. Not with abbreviations

Figure 5 row 235 OJIP curves...please write the entire name

Table 2. JIP please write the entire name

Table 3 and 4 looks different compared to table 2

Row 343-350 – It has been shown ...showed is repeated. The text could be reformulated in such a way not to repeat the same sentence structure and combine similar ideas

Row 353: The sentence is starting with an abbreviation again „PPH has been shown to be essential for chlorophyll catabolism during leaf brushing in a 353 variety of plants”

Row 386: „PIABS and PITOTAL” or (PIABS ) keep this abbreviations consistent in the entire text.

Row 400-401: please correct the text. Between rows 397-401 it is a single paragraph, please split the text into short sentences, the conclusions for each individual measured parameter or combinations between them in a logical manner.

The author contribution looks different compared to other manuscripts submitted to horticulture.

Reviewer 3 Report

Comments and Suggestions for Authors

I read with interest the manuscript entitled “End-of-day addition of a low intensity of a single light quality improves the resilience of cucumber seedlings under high light stress”. In this study, we used cucumber as a test material to investigate whether the addition of a single light quality before darkness could alleviate the high light stress suffered by cucumber, which would provide a reference for the cultivation of cucumber seedlings in greenhouses in the future. The subject of the article is important and has great relevance for the scientific environment of the study area. Therefore, the manuscript needs some adjustments so that it can then be forwarded to the publication process. The manuscript has the potential for publication in this journal Horticulturae and needs the following adjustments:

ABSTRACT

- The objective of the Abstract must be similar to that described in the Introduction.

- Describe the treatments and analyzes that were carried out. There is nothing talking about the Material and Methods section.

- Replace repeated keywords in the Title and add more terms.

INTRODUCTION

- It is not possible to state that light is the main abiotic stress for plants. Review this statement.

- Add some hypotheses before the objectives.

-

MATERIAL AND METHODS

- Describe in more detail how the MDA analysis was carried out.

- Quote the formulas to determine the chlorophyll content.

RESULTS

- Inform at the bottom of the Figures whether the comparisons are being made within each era.

- What does the X axis of each figure mean? Add title.

-

DISCUSSION

This section needs to be revised. There is a lot of unnecessary information and other information needs to be added, according to the best results.

Round 2

Reviewer 3 Report

Comments and Suggestions for Authors

Dear,

The authors were part of the previous suggestions.

They made no changes to the Discussion, as suggested. Therefore, I am submitting it again for another round of review.

Author Response

We have made some deletions from Discussion 4.1 and 4.2, and some additions to Discussion 4.3. Since there are fewer studies on the effects of different light quality treatments on the chlorophyll fluorescence fast kinetic curves of plants, there are fewer articles to refer to for discussion.